# Oleuropein as a Potent Compound against Neurological Complications Linked with COVID-19: A Computational Biology Approach

**DOI:** 10.3390/e24070881

**Published:** 2022-06-26

**Authors:** Talib Hussain, Alaa Hamed Habib, Misbahuddin M. Rafeeq, Ahmed Alafnan, El-Sayed Khafagy, Danish Iqbal, Qazi Mohammad Sajid Jamal, Rahamat Unissa, Dinesh C. Sharma, Afrasim Moin, Syed Mohd Danish Rizvi

**Affiliations:** 1Department of Pharmacology and Toxicology, College of Pharmacy, University of Hail, Hail P.O. Box 2440, Saudi Arabia; md.talib@uoh.edu.sa (T.H.); a.alafnan@uoh.edu.sa (A.A.); 2Department of Physiology, Faculty of Medicine, King Abdulaziz University, Jeddah 21589, Saudi Arabia; ahabib@kau.edu.sa; 3Department of Pharmacology, Faculty of Medicine, King Abdulaziz University, Jeddah 21589, Saudi Arabia; misbahuddinrafeeq@yahoo.com; 4Department of Pharmaceutics, College of Pharmacy, Prince Sattam Bin Abdulaziz University, Al-kharj 11942, Saudi Arabia; e.khafagy@psau.edu.sa; 5Department of Pharmaceutics and Industrial Pharmacy, Faculty of Pharmacy, Suez Canal University, Ismailia 41522, Egypt; 6Department of Medical Laboratory Sciences, College of Applied Medical Sciences, Majmaah University, Majmaah 11952, Saudi Arabia; da.mohammed@mu.edu.sa; 7Department of Health Informatics, College of Public Health and Health Informatics, Qassim University, Al Bukayriyah 52741, Saudi Arabia; m.quazi@qu.edu.sa; 8Department of Pharmaceutics, College of Pharmacy, University of Hail, Hail P.O. Box 2440, Saudi Arabia; srunissa@gmail.com; 9School of Life Sciences, The Glocal University, Saharanpur 247121, Uttar Pradesh, India; ddcsharma@gmail.com; 10Department of Microbiology, School of Life Sciences, Starex University, Gurugram 122413, Haryana, India

**Keywords:** oleuropein, COVID-19, SARS-CoV-2, TLR-4, 3CL^pro^, molecular dynamics

## Abstract

The association of COVID-19 with neurological complications is a well-known fact, and researchers are endeavoring to investigate the mechanistic perspectives behind it. SARS-CoV-2 can bind to Toll-like receptor 4 (TLR-4) that would eventually lead to α-synuclein aggregation in neurons and stimulation of neurodegeneration pathways. Olive leaves have been reported as a promising phytotherapy or co-therapy against COVID-19, and oleuropein is one of the major active components of olive leaves. In the current study, oleuropein was investigated against SARS-CoV-2 target (main protease 3CL^pro^), TLR-4 and Prolyl Oligopeptidases (POP), to explore oleuropein potency against the neurological complications associated with COVID-19. Docking experiments, docking validation, interaction analysis, and molecular dynamic simulation analysis were performed to provide insight into the binding pattern of oleuropein with the three target proteins. Interaction analysis revealed strong bonding between oleuropein and the active site amino acid residues of the target proteins. Results were further compared with positive control lopinavir (3CL^pro^), resatorvid (TLR-4), and berberine (POP). Moreover, molecular dynamic simulation was performed using YASARA structure tool, and AMBER14 force field was applied to examine an 100 ns trajectory run. For each target protein-oleuropein complex, RMSD, RoG, and total potential energy were estimated, and 400 snapshots were obtained after each 250 ps. Docking analyses showed binding energy as −7.8, −8.3, and −8.5 kcal/mol for oleuropein-3CL^pro^, oleuropein-TLR4, and oleuropein-POP interactions, respectively. Importantly, target protein-oleuropein complexes were stable during the 100 ns simulation run. However, an experimental in vitro study of the binding of oleuropein to the purified targets would be necessary to confirm the present study outcomes.

## 1. Introduction

The quest for COVID-19 medications began two years back, soon after SARS-CoV-2 origination was reported in Wuhan, China. Clinical findings on COVID-19 patients suggested the involvement of neurological, cardiovascular, and gastrointestinal systems along with the respiratory system [1]. Neurological complications are perhaps the most commonly associated manifestation of COVID-19 infection in patients after respiratory ailments [2,3]. Researchers want to trace the connecting link between neurological complications and COVID-19, and Toll-like receptor 4 (TLR4) gives off an impression of being one such link. Angiotensin Converting Enzyme 2 (ACE-2) is the main receptor for SARS-CoV-2 spike protein, although the spike protein can bind with equal competence to TLR-4 receptor as well [4,5]. The spike protein (SARS-CoV-2) binding with TLR-4 leads to cytokine storm generation that stimulates neuro-inflammation and neuro-degeneration in COVID infected patients [4,5,6,7]. Hence, TLR-4 could be used as a targeting protein for neurological complications associated with COVID-19 [4,8,9]. Recently, our team has reported dithymoquinone as a potent dual-targeting candidate against SARS-CoV-2 [10]. However, in the present study, the focus is on neurological complications associated with SARS-CoV-2 infection, and the compound selected is oleuropein. 

Olive leaves were reported as a potent phytotherapy for SARS-CoV-2 infection [11,12,13,14], and oleuropein is the major component of the olive leaf with SARS-CoV-2 inhibition potential [15]. Main protease SARS-CoV-2 enzyme (3CL^pro^), TLR-4 and Prolyl Oligopeptidases (POP) were used as target proteins for the present study. In fact, 3CL^pro^ cuts the viral protein at eleven different positions to create non-structural protein peptides important for viral replication [16,17,18]. Thus, new anti-COVID drug candidates could be designed using 3CL^pro^ as a target [16,18]. SARS-CoV-2 neuro-invasion enhances with increased ACE2 expression in neurons, astrocytes, and oligodendrocytes [19,20]. In addition, neurodegeneration, neuro-inflammation and alpha-synuclein aggregation in COVID-infected individuals are linked with SARS-CoV-2 potential to stimulate TLR-4 [4,5]. POP inhibition could reduce aggregation of alpha-synuclein in neurons [21]. Hence, investigating anti-COVID agent(s) against TLR-4 and POP might help in alleviating neurological complications linked with COVID-19. 

Oleuropein has been claimed to be a potent antiviral compound against respiratory syncytial virus, bovine rhinovirus, hepatitis virus, feline leukemia virus, herpes mononucleosis, para-influenza type 3 virus, rotavirus, and canine parvovirus [22,23]. Recently, oleuropein has shown strong binding with NSP15 endoribonuclease of SARS-CoV-2 [15]. Moreover, oleuropein is considered to be one of the strongest antioxidants identified in nature [24]. In this study, molecular docking and dynamic simulation approaches were applied to investigate the interaction and stability of oleuropein-target proteins (3CL^pro^, TLR-4 and POP) complex during 100 ns trajectory run.

## 2. Materials and Methods

### 2.1. Compounds and Target Protein Preparation

COCONUT online: Collection of Open Natural Products database [25] was used to retrieve different natural derivatives of oleuropein. Three-dimensional structure for oleuropein (CNP0209991), oleuropein aglycone (CNP0301195), Oleuropein-dialdehyde form (CNP0162125), and Oleuropein-aldehyde form (CNP0085322) were obtained in .pdb format. Lopinavir (ID: 92727), resatorvid (ID: 11703255) and berberine (ID: 2353) were retrieved from PubChem database. All these compound structures were further converted into pdbqt format using OpenBabel tool.

Target proteins 3CL^pro^ (PDB ID: 6LU7), TLR4 (PDB ID: 3FXI), and POP (PDB ID: 3DDU) were obtained from protein data bank. AutoDock 4.2 was used to add polar hydrogen, solvation parameters, and Kollman united atom charges on the target proteins and convert them into pdbqt format [26]. 

### 2.2. Molecular Docking and Interaction Analysis

AutoDock Vina was used for docking of each compound with the target proteins (3CL^pro^, TLR4, and POP) [27]. Proteins were targeted by setting the grid co-ordinates towards the active site. Grid box center was kept as x: −16.539; y: 15.246; z: 67.334 for 3CL^pro^, x: 9.261; y: 0.905; z: 20.315 for TLR4, and x: −8.263; y: 14.166; z: 27.480 for POP, respectively. However, size of grid box was kept as 60 × 60 × 60. The results were obtained in terms of affinity (kcal/mol) and each docking interaction was divided into 10 modes of descending order. The best confirmation was saved by using Pymol. The interaction detail for each docking complex was analyzed by Discovery Studio Visualizer.

### 2.3. Molecular Dynamics Simulation Analysis

Based on docking results and comparison with positive control, oleuropein (CNP0209991) complex with 3CL^pro^, TLR4, and POP were selected for molecular dynamic simulation by using YASARA-structure tool [28]. AMBER14 force field was used to set periodic boundary (20 Å) of the simulation cell and water as a solvent was filled with density of 0.998 g/mL. H-bond network was optimized to augment the solute stability, and protein protonated state was tuned at pH 7.4 via pKa prediction [29]. System environment was neutralized by adding sodium chloride ions, and system energy minimization was done by YASARA-structure tool. Simulation annealing and steepest descent approaches were applied to resolve clatters before running a 100 ns trajectory by AM1BCC [30] and GAFF2 [31] for oleuropein, TIP3P for water, and AMBER14 force field for solute [32]. For van der Waals forces, an 8 Å cut-off was applied, whereas no cut-off was applied for electrostatic forces and Particle Mesh Ewald algorithm. NPT ensembles were applied to perform position restraining to equilibrate the system [33]. All the protein-ligand complex were equilibrated for non-bonded and bonded interactions with 2.5 and 5.0 fs multi-time step at 298K temperature at pressure of 1bar. NPT ensemble iso-baric and -thermal environments were maintained throughout the simulation run, and Berendsen thermostat time average -pressure and -temperature approach was used to maintain pressure and temperature of the system [34]. However, multi-step algorithm through modified LINCS version [35] was used to constrain the angles and bond. It is noteworthy to mention that simulation process was completed by using user-friendly macros ‘md_runfast.mcr’ YASARA-structure tool, and trajectory analysis was performed by ‘md_analyze.mcr’. In total, 400 different snapshots were taken after every 250 ps to create figures using YASARA-structure tool.

## 3. Results and Discussion

Multi-organ complications and pathological changes are normal in COVID-19 patients [1]. The most noticeable issue is the neuro-pathophysiological state of the patient after respiratory distress [2,3]. TLR-4 and POP appears to be a connection for neurological complications linked to COVID-19 [4,5]. In the present study, a promising anti-COVID compound oleuropein from olive leaves [11,12,13,14,15] was selected to predict its potency in alleviating the neurological impact of COVID-19 infection by targeting oleuropein against TLR-4 and POP.

### 3.1. Molecular Docking Analysis

Initially, COCONUT online database was used to retrieve different derivatives of oleuropein, i.e., oleuropein (CNP0209991), oleuropein aglycone (CNP0301195), oleuropein-dialdehyde form (CNP0162125), and oleuropein-aldehyde form (CNP0085322). All these compounds with positive control were docked with the target proteins 3CL^pro^, TLR-4 and POP by using AutoDock Vina (Table 1). Oleuropein (CNP0209991) showed better binding energy than other derivatives against all the target proteins. Gibbs free energy (ΔG) of oleuropein interaction with 3CL^pro^ was −7.8 kcal/mol, whereas lopinavir (control) showed ΔG as −8.4 kcal/mol. Recently, Selleckchem Natural Product Database [15] has been screened against non-structural protein (NSP15) of SARS-CoV-2, and oleuropein predicted to be the second-best compound in terms of interaction energy, i.e., −8.5 kcal/mol against NSP15. In addition, oleuropein was claimed to be the most potent anti-viral compound against different human viruses [22,23]. However, our study intended to explore a different aspect of oleuropein, i.e., as an anti-COVID agent that can mitigate the neurological effects associated with COVID-19. Thus, TLR-4 and POP were targeted with oleuropein. Docking interaction of oleuropein with TLR-4 showed ΔG as −8.3 kcal/mol, whereas positive control (resatorvid) showed ΔG as −7.1 kcal/mol. On the other hand, ΔG of oleuropein interaction with POP was −8.5 kcal/mol, which was better than berberine interaction with POP (−8.1 kcal/mol). The results predicted that oleuropein interaction was better than the positive control against TLR-4 and POP. Moreover, to gain a deeper insight into oleuropein interaction with target proteins Discovery Studio Visualizer tool was applied, and re-docking validation experiments were also executed.

### 3.2. Interaction Analysis of Oleuropein with Target Proteins

Oleuropein (CNP0209991) interacted with the active site of 3CL^pro^ (Figure 1). However, the protocol was standardized by re-docking with native ligand (N3) [36] and superimposition of re-docked native ligand and native ligand confirmed the standardization (Figure 1B). The control compound lopinavir was also docked within the same active site cavity of 3CL^pro^. Interestingly, oleuropein, native ligand, and lopinavir interacted at a similar position of the active site of 3CL^pro^ (Figure 1C). Oleuropein showed strong hydrogen, hydrophobic, and van der Waals interactions with 3CL^pro^ (Figure 1D). A total nine hydrogen bonds were observed, of which eight were conventional hydrogen bonds with THR26, SER46, SER144, MET165, GLU166, ARG188, THR190, GLN192 and one was carbon hydrogen bond with PHE140. MET 165 was also involved in pi-alkyl interaction, whereas MET49 was involved in alkyl interaction. In addition, twelve amino acids of active site cavity of 3CL^pro^ were involved in van der Waals interactions with oleuropein. In comparison, the positive control lopinavir showed one hydrogen bonding with GLN189 and one pi-sulphur interaction with MET165, whereas 14 amino acid residues showed van der Waals interactions with 3CL^pro^ (Figure 1E). 3CL^pro^ or M^pro^ has a large active site with five sub-pockets (S1–S5) [37]. Importantly, oleuropein has shown a strong hydrogen bonding with all these sub-pockets via THR26 (S2 catalytic center and S5), SER46 (S5), SER144 (S1), MET 165 (S1, S3, and S4), GLU166 (S1 and S4), ARG188 (S3 and S4), THR190 (S4), GLN192 (S4), and PHE 140 (S1). THR26 is part of the S2 catalytic center of 3CL^pro^ and has been reported along with GLU166 and GLN189 as a crucial amino acid for 3CL^pro^ targeting [10,38,39]. Even native ligand re-docking also suggested the involvement of GLU166, THR190, and PHE140 amino acid residues in hydrogen bonding. Thus, the findings with oleuropein are quite encouraging for 3CL^pro^ targeting.

TLR-4 and Myeloid differentiation factor 2 (MD-2) binding pocket was the focus of the current study. Lipopolysaccharide MD-2 is a co-receptor for TLR-4 [40], and activation of TLR-4 via SARS-CoV-2 spike protein is also connected with MD-2 [41]. Hence, restricting TLR-4 and MD-2 binding pockets could play an important role in alleviating neuro-inflammation linked with COVID-19 infection. Oleuropein (CNP0209991) and resatorvid (control compound) were subjected to docking with the crucial binding pocket of TLR-4 and MD-2 (Figure 2). Here, protocol standardization was performed by re-docking native ligands (6 lipopolysaccharide chains) with the TLR-4 binding pocket (Figure 2E–J). Figure 2A,B illustrate the superimposition of oleuropein, native ligands and resatorvid. Superimposition revealed that all the ligands, oleuropein, and resatorvid interacted within the same vicinity of the TLR-4 binding pocket. In addition, further investigation of the interaction was performed for oleuropein and resatorvid (Figure 2C,D). Phenylalanine amino acid residues, i.e., PHE119, PHE126, and PHE 151 of the TLR-4-MD-2 binding pocket played a crucial role in binding oleuropein (Figure 2C). PHE126 showed hydrogen bonding, whereas PHE119 and PHE151 showed pi-sigma and pi-pi stacking interactions. Thirteen amino acids were bonded with oleuropein through van der Waals interactions. In comparison, resatorvid showed hydrogen bond with CYS133, pi-alkyl interactions with PHE76, PHE151, ILE153, and alkyl interactions with ILE32, LEU78, VAL135. PHE126 plays a crucial role in bridging between TLR4 and MD-2 and expected to form a molecular switch for endotoxic signaling and dimer formation [42,43,44]. Importantly, oleuropein showed a strong hydrogen bonding with PHE126. In a previous study, a chalcone derivative competitively displaced lipopolysaccharides from the TLR-4 and MD-2 binding pocket via binding with PHE119 and PHE151 [45]. Thus, it can be stated that oleuropein has the ability to block the TLR-4 binding pocket efficaciously. 

Inflammation and α-synuclein aggregation in neurons could be triggered by POP [46,47]. Further, POP also contributes to COVID-19-linked neuro-inflammation, and acts as a determining factor for angiotensin 2 levels in COVID-19-infected patients [48]. Therefore, targeting POP might alleviate the neuro-suffering of COVID-19 patients. In the present study, oleuropein (CNP0209991) and berberine (control compound) was docked with the POP (Figure 3). To validate the protocol native ligand GSK552 was redocked. However, the superimposed image (Figure 3B) showed that berberine and oleuropein interacted to the different site from the native ligand. Interaction analysis of oleuropein with POP revealed ten hydrogen bondings via TYR71, TYR73, ASN91, ASN96, ARG98, GLN397, THR399, GLY405, TYR484, THR686, two alkyl interactions with LYS75 and VAL425, and one pi-alkyl interaction with VAL383. In addition, seven amino acid residues of POP were involved in van der Waals interactions. In contrast, berberine interacted with POP through two hydrogen bonds (TYR76, GLY405), one pi-pi T-shaped (TYR71), and three alkyl (LYS75, VAL689, VAL693) interactions, whereas, nine amino acids were involved in van der Waals interactions. POP catalytic triad is formed by HIS680, ASP641, and SER554, whereas it has three specificity pockets at the active site consisting of PHE476, ASN555, VAL580, TRP595, TYR599, VAL644, ARG643, PHE173, MET235, CYS255, ILE591, and ALA594 [49]. Oleuropein showed a strong hydrogen bonding with POP but did not interact with any of these important amino acid residues of the active site. Nevertheless, phosphorylation of POP takes place at TYR71 amino acid residue [50,51], and binding of oleuropein to TYR71 might inhibit its phosphorylation.

### 3.3. Molecular Dynamic (MD) Simulation Analysis

Molecular docking analysis revealed that oleuropein has a strong inhibitory potential against the target proteins, especially for 3CL^pro^ and TLR-4 receptor. Hence, oleuropein was subjected to dynamic simulation analysis to evaluate the stability of oleuropein-target protein complexes during 100 ns trajectory run. YASARA-structure tool was applied to perform simulation of the oleuropein-target protein complex. For each target protein complex, simulation parameters were maintained and optimized during the 100 ns run. The main objective of the simulation was to understand the stability and binding affinity of each complex.

Total potential energy (TPE) plots were created by applying AMBER14 force field (Figure 4). Once the simulation begins from ground-zero/ frozen-energy minimization state, a rapid increase in energy is typically observed during the initial few picoseconds of simulation time. The energy increase is due to storage of kinetic energy as potential energy. In addition, potent energy is normally not reduced because of counter ions during the large timescale. In fact, they are arranged close to the charged solute groups with lower potential energy, and eventually dispersed to increase the potential energy and entropy of the system. Oleuropein and 3CL^pro^ complex TPE plot (Figure 4A) showed fluctuations from −1,217,000 to −1,222,500 kJ/mol, whereas TPE plot for oleuropein and TLR4 (Figure 4B) showed fluctuations between −1,89,8000 and −1,90,8000 kJ/mol. Oleuropein-POP complex TPE plot (Figure 4C) indicated the variability from −1,465,000 to −1,472,000 kJ/mol.

Root mean square deviations (RMSD) for Oleuropein-3CL^pro^, Oleuropein-TLR4, and Oleuropein-POP complex were shown in Figure 5. It contains backbone RMSD as pink color, CαRMSD as green color, and all heavy atoms RMSD as blue color. Oleuropein and 3CL^pro^ complex showed overlapping of backbone and Cα RMSD during the entire 100 ns run (Figure 5A). Minimal fluctuations can be observed from 38 to 62 ns and 85 to 100 ns. Overall, the fluctuations were in the range of 1.5 to 3 Å, which is well within the acceptable range of 2 Å. Figure 5B displays the RMSD of Oleuropein-TLR-4 complex. Here, all RMSDs were overlapping with minimal fluctuations, and the range of fluctuations were between 2.5 to 4.0 Å during the entire 100 ns run. However, Oleuropein-POP complex showed complete overlapping from 12 ns till the end of 100 ns run (Figure 5C). Fluctuation was 2.5 to 6 Å starting from 30 ns until 45 ns, then regained some stability from 45 ns to 60 ns, although it began to fluctuate again from 60 ns to 80 ns. Oleuropein-3CL^pro^ and Oleuropein-TLR-4 complexes were comparatively more stable than Oleuropein-POP complex.

Radius of Gyration (RoG) of all three complexes during the 100 ns run is presented in Figure 6. RoG stability correlates to the protein stability during the simulation run, and its evaluation relies on the center of mass of the protein which characterizes the structure compactness during the simulation. Oleuropein and 3CL^pro^ complex RoG plot represents minimal fluctuations 22.3 to 22.5 Å during the initial 33 ns, after which fluctuations increased up to 22.9 Å until the end of the 100 ns run (Figure 6A). Overall, fluctuation for the entire 100 ns was 0.5 Å for Oleuropein-3CL^pro^ complex. RoG plot for oleuropein and TLR-4 complex showed fluctuations 29.2 to 31.6 Å during the 100 ns run trajectory (Figure 6B). In fact, fluctuations increased for Oleuropein-TLR-4 complex in comparison to Oleuropein-TLR-4 complex, whereas the RoG plot of Oleuropein-POP complex showed minimal fluctuations (26.4 to 26.8 Å) from 30 ns to 100 ns (Figure 6C). However, average fluctuations were within the acceptable range of 0.5 Å, 1.4 Å, and 0.3 Å for Oleuropein-3CL^pro^, Oleuropein-TLR-4 and Oleuropein-POP complex, respectively.

Figure 7 presents snapshots of the simulation during different time intervals. The black arrow has been used to locate oleuropein in the complex form with the respective target proteins, i.e., 3CL^pro^ (Figure 7A), TLR-4 (Figure 7B) and POP (Figure 7C). Importantly, oleuropein remained bonded to all the target proteins in a stable conformation during the entire 100ns run. Overall, the dynamic simulation results and interaction analysis revealed that oleuropein can form stable complexes with all the target proteins. TLR-4 is a potential target for different neurological disorders, and TLR-4 inhibitors have shown suppression of neuroinflammation via reducing the production of neuroinflammatory mediators [52]. There are several clinical trials underway to develop TLR-4 inhibitors/antagonists for various applications such as neuropathic pain, traumatic brain injury, rheumatoid arthritis, diabetes, myeloid leukemia, among others [53]. On the other hand, POP plays a key role in various pathological neurodegenerative processes, which subsequently accelerate the development of POP inhibitors. Successful preclinical trials have demonstrated the potency of POP inhibitors in reversing the memory loss caused by neurological disorders [54]. Thus, oleuropein stable binding with TLR-4 and POP will not only limit neuro-COVID effects but pave the way for treatment strategies against other neurological disorders as well. Oleuropein is known for its antiviral activity and reported to inhibit fusion and infectivity of viruses [22,23,55,56]. In 2019, phenolic compounds of virgin olive oil showed TLR4/NLRP3 modulating potential to curb inflammatory response on brain cells [57]. Recently, oleuropein potency has been reported against SARS-CoV-2 non-structural protein 15, and some promising results were obtained [15]. In addition, olive leaves (whose major constituent is oleuropein) have been used as a co-therapy for COVID-19 patients [11,12,13,14]. It is noteworthy to mention that before selecting oleuropein for the present study the authors have explored dithymoquinone as a potent multi-target candidate against SARS-CoV-2 [10] and achieved some worthy outcomes on dithymoquinone analogues designed against neuro-COVID [unpublished data]. The neurological impact of COVID-19 is a well-established fact. Thus, anti-COVID medication with the ability to ease neurological complications appears to be an appealing and desirable strategy. Experimental in vitro and in vivo studies are needed to confirm oleuropein as a potential neuro-COVID candidate. However, it has been observed that virtual computational findings often correlate well with real outcomes. Nevertheless, the preliminary outcomes of the present investigation have established a base for further exploration of oleuropein as a future neuro-COVID therapeutic agent.

## 4. Conclusions

The rise in neurological complications in SARS-CoV-2 infected patients has prompted researchers to ascertain the reason behind it. SARS-CoV-2 spike protein binding affinity towards TLR-4 provided a clue regarding the cytokine storm and its linked neuro-inflammation. In the present study, oleuropein was selected (based on strong antiviral potential) to evaluate its potency against SARS-CoV-2 main protease (3CL^pro^) and SARS-CoV-2 associated neuro-targets (TLR4 and POP). The results revealed that oleuropein showed strong interaction with 3CL^pro^, TLR4, and POP which were comparable with the positive control. Oleuropein interacted within the same active site of 3CL^pro^ and TLR4 as native ligands and control compounds, whereas oleuropein interaction with POP was somewhat offsite compared with native ligand but within the site of interaction of the control compound. Molecular dynamic results showed that all the complexes were stable during the 100 ns run trajectory. Oleuropein-3CL^pro^ and Oleuropein-TLR-4 complexes were comparatively more stable than Oleuropein-POP complex with minimal fluctuations in RMSD plots. RoG plot revealed that 3CL^pro^ and POP protein structures were more compact than TLR-4 structure during the simulation run, however, fluctuations for all the complexes were under the acceptable limit. The concept of ‘an anti-COVID candidate drug that can mitigate neurological complications in COVID-19-infected patients’ is like ‘filling two needs with one deed’. The present investigation suggests oleuropein as a potential candidate that can target SARS-CoV-2 and alleviate neurological manifestations associated with it at the same time. This might pave the way to establishing a potent candidate against neuro-COVID complications. However, in vitro experimental validation is warranted before confirming the present study outcomes. 

## Figures and Tables

**Figure 1 entropy-24-00881-f001:**
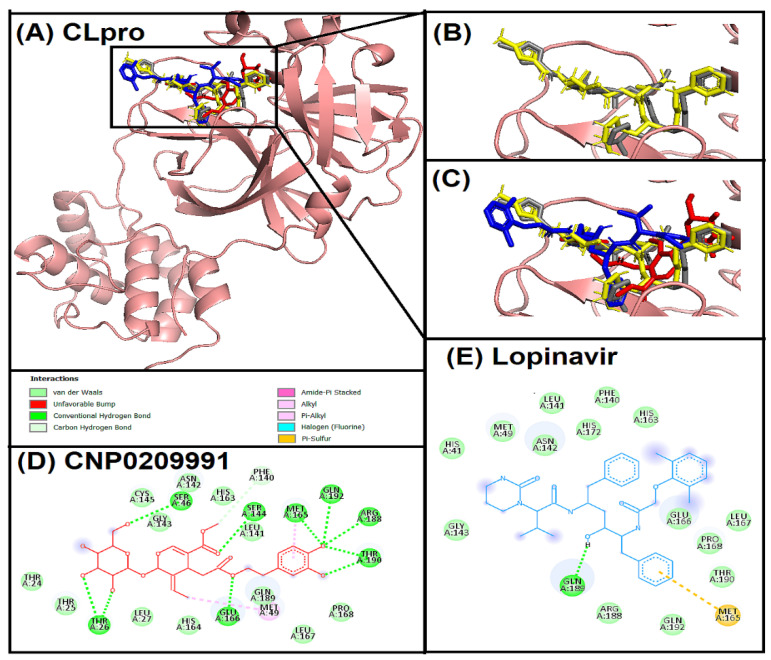
Superimposed image of the active site cavity of 3CL^pro^ (PDB ID: 6LU7) after docking. (**A**) All the docked ligands (native ligand: Grey color; Redocked ligand: Yellow color; Oleuropein (CNP0209991): Red color; Lopinavir: Blue color) in the catalytic active site. (**B**) Superimposed magnified image of native ligand and redocked native ligand. (**C**) Magnified image of all the docked ligands. (**D**) Molecular interaction analysis of oleuropein with 3CL^pro^ amino acid residues. (**E**) Molecular interaction analysis of lopinavir with 3CL^pro^ amino acid residues.

**Figure 2 entropy-24-00881-f002:**
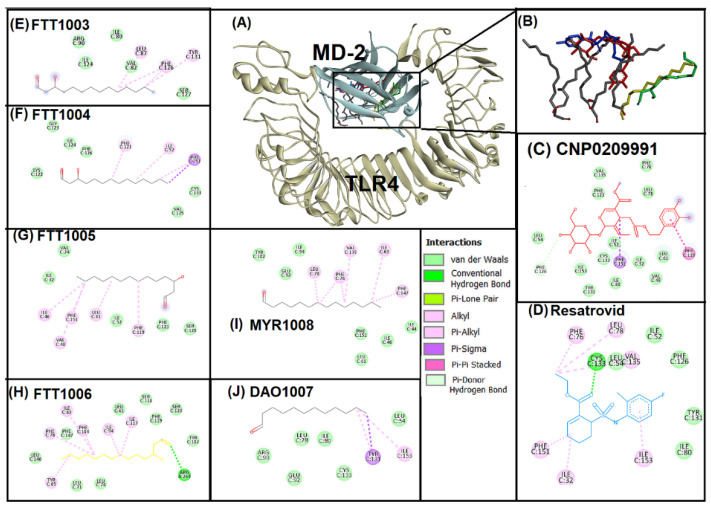
Superimposed image of docked ligands in the active site of TLR4 (PDB ID: 3FXI). (**A**) All the docked ligands (Native ligand: Grey color; Redocked ligand: Yellow and Green color; Oleuropein (CNP0209991): Red color; Resatorvid: Blue color) in the catalytic active site. (**B**) Magnified image of all the docked ligands. (**C**) Molecular interaction analysis of oleuropein (CNP0209991) with amino acid residues. (**D**) Molecular interaction analysis of resatorvid with amino acid residues. (**E**–**J**) Molecular interaction analysis of six lipid chains of lipopolysaccharides with amino acid residues.

**Figure 3 entropy-24-00881-f003:**
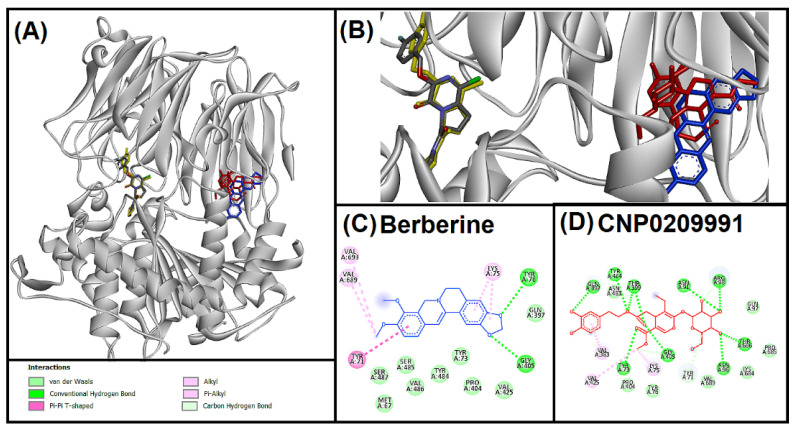
Superimposed image of docked ligands in the active site of POP (PDB ID: 3DDU). (**A**) All the docked ligands (native ligand: Grey color; Redocked ligand: Yellow color; Oleuropein (CNP0209991): Red color; Berberine: Blue color). (**B**) Superimposed magnified image of all the docked ligands. (**C**) Molecular interaction analysis of berberine with amino acid residues. (**D**) Molecular interaction analysis of oleuropein (CNP0209991) with amino acid residues.

**Figure 4 entropy-24-00881-f004:**
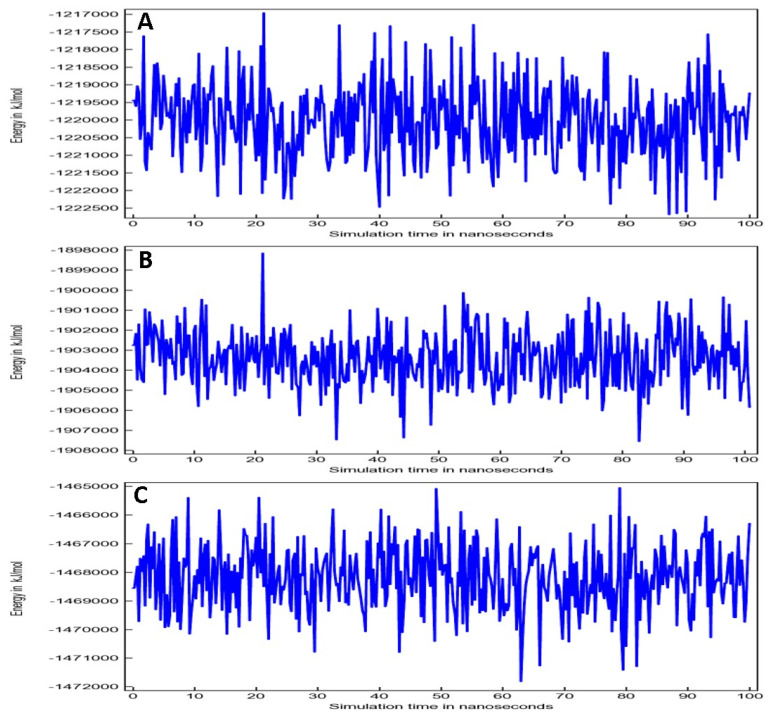
Total potential energy for (**A**) Oleuropein-3CL^pro^ complex, (**B**) Oleuropein-TLR4 complex and (**C**) Oleuropein-POP complex.

**Figure 5 entropy-24-00881-f005:**
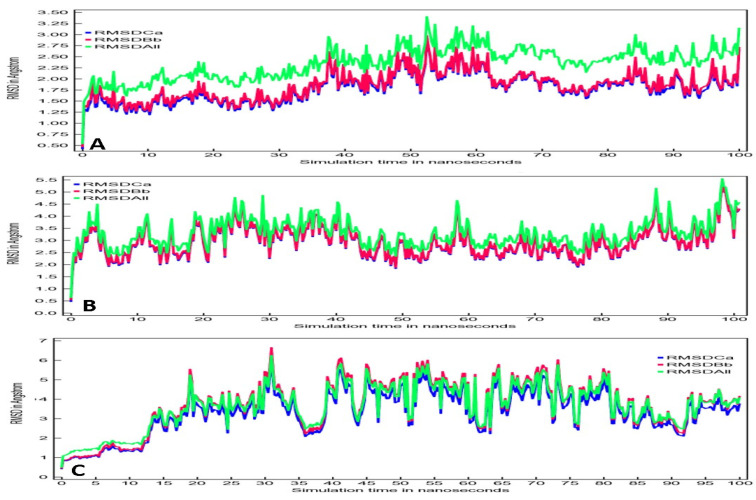
RMSD plots for (**A**) Oleuropein-3CL^pro^ complex, (**B**) Oleuropein-TLR4 complex, and (**C**) Oleuropein-POP complex.

**Figure 6 entropy-24-00881-f006:**
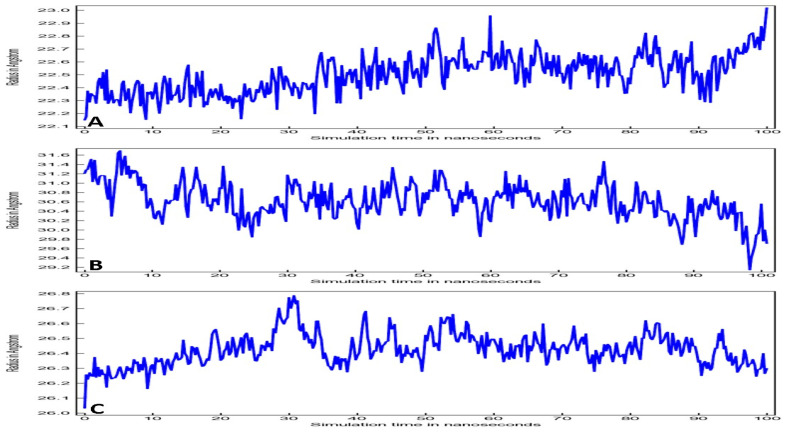
RoG plots for (**A**) Oleuropein-3CL^pro^ complex, (**B**) Oleuropein-TLR4 complex, and (**C**) Oleuropein-POP complex.

**Figure 7 entropy-24-00881-f007:**
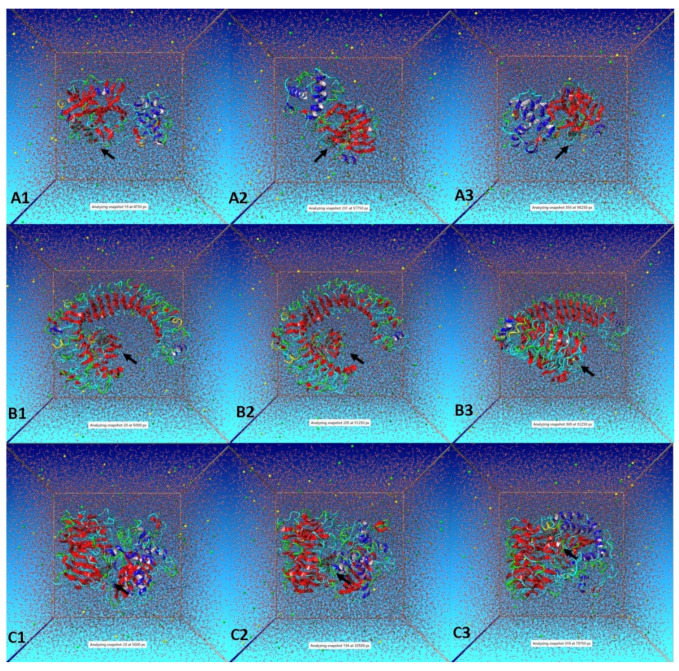
Snapshots of simulation runs for (**A**) Oleuropein-3CL^pro^ complex, (**B**) Oleuropein-TLR4 complex, and (**C**) Oleuropein-POP complex. Here, black arrow indicates the location of Oleuropein.

**Table 1 entropy-24-00881-t001:** Molecular docking results of oleuropein derivatives and control compounds’ interaction with different targets.

Compounds	3CL^pro^	TLR-4	POP
CNP0085322	−6.7 kcal/mol	−7.0 kcal/mol	−7.4 kcal/mol
CNP0162125	−6.2 kcal/mol	−6.7 kcal/mol	−6.5 kcal/mol
CNP0209991	−7.8 kcal/mol	−8.3 kcal/mol	−8.5 kcal/mol
CNP0301195	−6.7 kcal/mol	−7.4 kcal/mol	−6.8 kcal/mol
Lopinavir	−8.4 kcal/mol	-	-
Resatorvid	-	−7.1 kcal/mol	-
Berberine	-	-	−8.1 kcal/mol

## Data Availability

Not applicable.

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
