# Peer review of "Oleuropein as a Potent Compound against Neurological Complications Linked with COVID-19: A Computational Biology Approach"

_entropy, 2022, doi:10.3390/e24070881_

Round 1
Reviewer 1 Report
In this manuscript authors investigate the neuroprotective effects of oleuropein against neurological complications that accompany COVID-19.
They explore the binding pattern of oleuropein with three target proteins: main protease (3CLPRO), TLR4 and Propyl Oligopeptidases. By docking analysis, the authors demonstrate stable binding for oleuropein with 3CLPRO, TLR4 and POP, suggesting that oleuropein scaffold can be used to design anti-COVID agents to relieve neurological signs associated with COVID-19.
Although other previous virtual studies have selected (by molecular docking and molecular dynamics) several phenolic compounds for their anti-SARS-Cov-2 activity, specifically against hyperinflammatory response (see Front. Mol. Biosci. 7:627767. doi: 10.3389/fmolb.2020.627767), the results of the present paper are interesting, well organized and the manuscript is clear written. Moreover, the methods are appropriate.
The novelty of this paper concerns the binding studies between oleuropein and TLR4, a receptor expressed by all main CNS cell types and which role is critical for the CSN inflammatory response, and 3CLPRO protease and POP. Whether neurological manifestations in patients with COVID-19 are due to direct invasion of the virus or result from SARS-CoV-2-dependent neuroinflammatory response remains speculative.
Oleuropein shares similarities in their chemical structure with the most phenolic compounds present in olive leaves and not in other edible fruit derivates. Importantly, the anti-neuroinflammatory action of many phenolic compounds, including oleuropein, is own through the activation of the TLR4/NLRP3 inflammasome axis (see Molecules 2019, 24(24):4523. doi: 10.3390/molecules24244523).
To date, exhaustive study showing the physical interaction between oleuropein, TLR4, 3CLPRO, and POP was missing. Therefore, the present work deserves to be taken into account.
Minor spell check is required.
Author Response
Reviewer 1:
- Minor spell check is required.
- First of all, we would like to thank the honorable reviewer for all the appreciable words. We have duly added the two references (Front. Mol. Biosci. 7:627767. doi: 10.3389/fmolb.2020.627767 and Molecules 2019, 24(24):4523. doi: 10.3390/molecules24244523) to deliver the concept in a better way. However, entire MS has been checked for the spelling and grammatical errors. Changes are highlighted in yellow.

Reviewer 2 Report
The paper “Oleuropein as a potent compound against neurological compli-2 cations linked with CoVID-19: a computational biology approach” by Hussain et al. describes a molecular dynamics study of the potential binding of oleuropein, a natural compound present in olive leaves, to some targets of CoVID-19. The particularly relevant aspect of this study is that, if oleuropein would present beneficial effects on the neurological complications related to the long-Covid pathology, the compound does not need any specific authorisation, since the compound is already sold as a major component of a supplement freely available on the market (it is contained, for example, in a supplement called Eumetab, suggested to prevent the type-2 diabetes, in addition to other suggested activities- antioxidant, nutraceutical, apoptosis inducer etc).
The docking of the compound to different possible targets was performed using control compounds (resatrovid, berberin, Lopinavir) and, as far as I can state not being a real expert of molecular dynamics, the dynamics were run appropriately.
There are in my opinion two limitations of this approach.
The first is that the link of the protein targets with the neurological effects of COVID-19 pathologies are quite weak. TLR-4 and POP are present in humans independently from the COVID infection. Which would be the effect of the binding of oleuropein to TLR.4 and POP on healthy people? Perhaps some comments on this aspect should be added.
The second (and stronger) limitation of the paper is the lack of experimental data on the effective binding of the compound. It is in fact stated at the end of the abstract and in conclusions that “The findings suggested that oleuropein scaffold could be used …”. This fact could perhaps be better stressed adding that “… an experimental in vitro study of the binding of oleuropein to the purified targets would be necessary to confirm the binding” (or something similar).
Minor points:
Fig.7 is aesthetically nice, but very hard to interpret. Perhaps in the pdf I received the resolution is low and the original is better?
Author Response
Reviewer 2:
First of all, I would like to thank the honorable reviewer for giving precious time from his busy schedule to read the MS and provide us comments that can improve the quality of our research article. All the comments have been addressed and highlighted in Yellow.
1. The first is that the link of the protein targets with the neurological effects of COVID-19 pathologies are quite weak. TLR-4 and POP are present in humans independently from the COVID infection. Which would be the effect of the binding of oleuropein to TLR.4 and POP on healthy people? Perhaps some comments on this aspect should be added.
- As per the suggestion of the honorable reviewer, some discussion on this aspect has been duly added in the revised version of the MS.
2. The second (and stronger) limitation of the paper is the lack of experimental data on the effective binding of the compound. It is in fact stated at the end of the abstract and in conclusions that “The findings suggested that oleuropein scaffold could be used …”. This fact could perhaps be better stressed adding that “… an experimental in vitro study of the binding of oleuropein to the purified targets would be necessary to confirm the binding” (or something similar).
- As per the suggestion of the reviewer, the statement has been duly modified in the Abstract and Conclusion section.
3. Fig.7 is aesthetically nice, but very hard to interpret. Perhaps in the pdf I received the resolution is low and the original is better?
- Figure 7 are the actual snapshots taken during the simulation process without any modification. However, we have tried to duly increase the resolution of the Figure 7 (from 150 dpi to 600 dpi) and added to the revised version of the MS.

Round 2
Reviewer 2 Report
The authors have satisfactory modified the text and in my opinion the paper can now be published.